# FLAIR: Federated Learning Annotated Image Repository

**Congzheng Song**[*]
Apple
csong4@apple.com

**Filip Granqvist**[*]
Apple
fgranqvist@apple.com

**Kunal Talwar**
Apple
ktalwar@apple.com

## Abstract

Cross-device federated learning is an emerging machine learning (ML) paradigm where a large population of devices collectively train an ML model while the data remains on the devices. This research field has a unique set of practical challenges, and to systematically make advances, new datasets curated to be compatible with this paradigm are needed. Existing federated learning benchmarks in the image domain do not accurately capture the scale and heterogeneity of many real-world use cases. We introduce FLAIR, a challenging large-scale annotated image dataset for multi-label classification suitable for federated learning. FLAIR has 429,078 images from 51,414 Flickr users and captures many of the intricacies typically encountered in federated learning, such as heterogeneous user data and a long-tailed label distribution. We implement multiple baselines in different learning setups for different tasks on this dataset. We believe FLAIR can serve as a challenging benchmark for advancing the state-of-the art in federated learning. Dataset access and the code for the benchmark are available at https://github.com/apple/ml-flair.

## 1 Introduction

Remote devices connected to the internet, such as mobile phones, can capture data about their environment. Machine learning algorithms trained on such data can help improve user experience on these devices. However, it is often infeasible to upload this data to servers because of privacy, bandwidth, or other concerns.

Federated learning [40] has been proposed as an approach to collaboratively train a machine learning model with coordination by a central server while keeping all the training data on device. Coupled with differential privacy, it can allow learning of a model with strong privacy guarantees. Models trained via private federated learning have successfully improved existing on-device applications while preserving users' privacy [18, 19, 41].

This has led to many ongoing research on designing better algorithms for federated learning applications. Centralized (non-federated) machine learning has benefited tremendously from standardized datasets and benchmarks, such as Imagenet [10]. To evaluate and accelerate progress in (private) federated learning research, the community needs similarly high quality large-scale datasets, with benchmarks. Ideally, the dataset would be representative of the challenges identified as important by the community [28]. Additionally, the benchmark should provide common, agreed-upon metrics to allow comparison of privacy, utility, and efficiency of various approaches.

Federated data may have various non-IID characteristics that are seldom encountered in traditional ML [28]. These include shifts in feature and label distribution, imbalanced user dataset sizes, drift in feature distribution conditioned on the labels and shift in the labeling function itself. This is caused

---

[*]Equal contribution

36th Conference on Neural Information Processing Systems (NeurIPS 2022) Track on Datasets and Benchmarks.

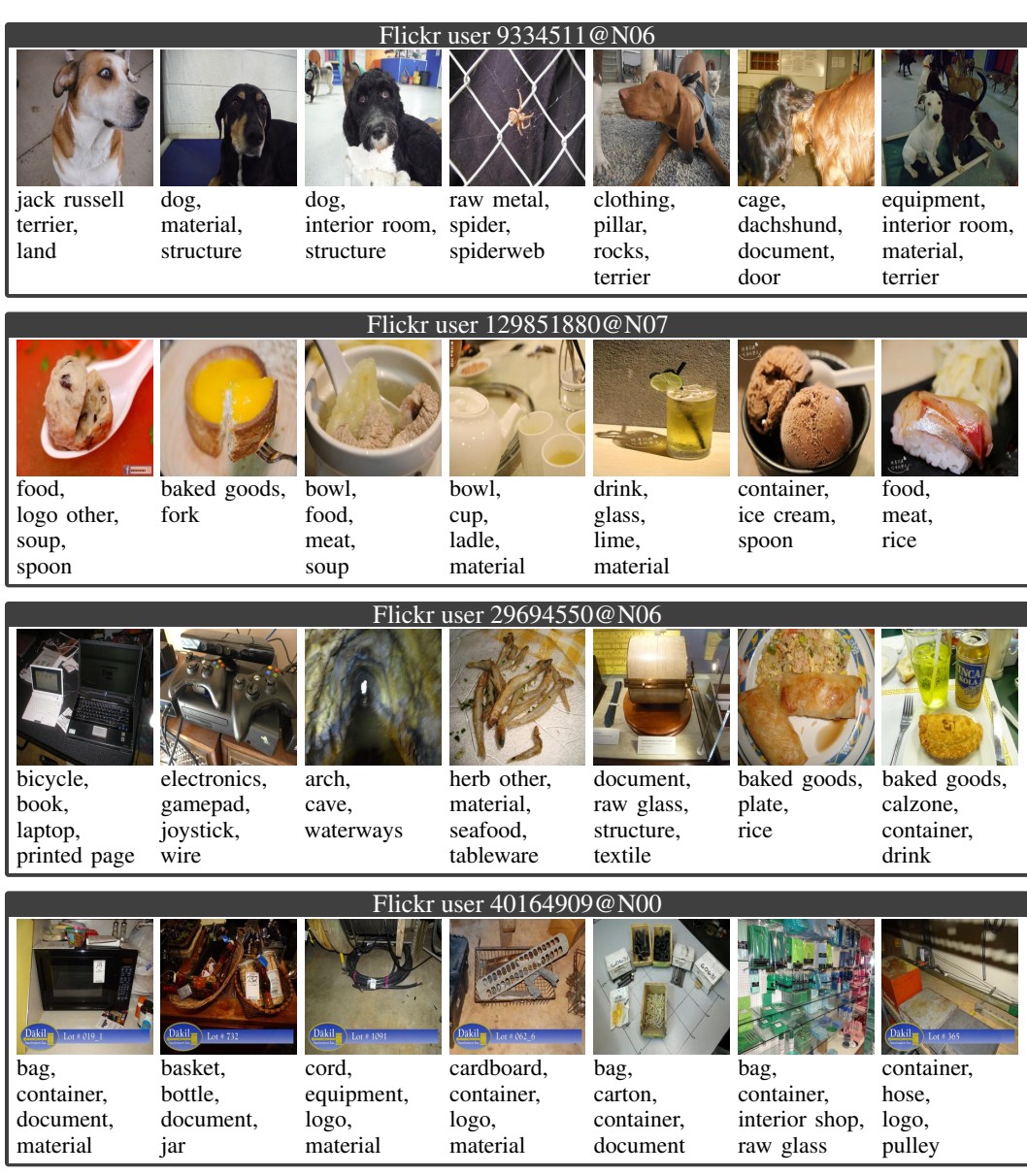

Figure 1: FLAIR sample images and labels. Images in the same row are from the same Flickr user. Captions below each image are the annotated fine-grained labels.

by the independent and diverse user-specific contexts that predicate the data generation process. For example, the style and content of a written message may differ depending on the author's age, culture, and geographical location. Indeed such heterogeneity can be seen in text datasets commonly used as benchmarks (see Section 2).

However the image domain suffers from a limited selection of large-scale datasets with realistic user partitions to benchmark algorithms and models (see Section 2). When new hypotheses are tested, researchers typically use centrally available data to simulate the federated setting. For example, many works are evaluated by repurposing traditional benchmarks, such as MNIST [35] and CIFAR10 [32], by creating artificial user partitions [24]. It is unclear if such artificial partitions are realistic enough to give confidence that hypotheses evaluated on these will transfer to federated learning in a real-world scenario.

We introduce FLAIR, a large-scale multi-label image classification dataset, for benchmarking federated learning algorithms and models. The dataset has a total of 429,078 images originating from

Flickr [1] and partitioned by 51,414 real user IDs. The images are annotated with labels from a two-level hierarchy, allowing us to define benchmarks with two levels of difficulty: the easier task has 17 coarse-grained classes and the harder task has 1,628 fine-grained classes. FLAIR also inherits many of the aforementioned non-IID characteristics:

- Imbalanced partitions — Users have different number of images. The majority of users have only 1-10 images, but the most active users have hundreds of images.
- Feature distribution skew — Users have different cameras, camera settings, which affect pixel generation.
- Label distribution skew — Users take photos of objects that align with their interests, which vary across photographers.
- Conditional feature distribution skew — Photos of the same category of objects can look very different due to weather conditions, cultural and geographical differences.

We provide benchmarks and analyze the performance of different settings of interest for FLAIR: centralized learning; federated learning; federated learning with central differential privacy; using random initialization of model parameters; and using model parameters pretrained on ImageNet [45].

## 2 Related Work

Previous work has mainly used two methods for preparing federated datasets: artificial partitioning of existing open-source datasets not originally purposed for federated learning [24], and constructing realistic partitions using real user identifiers preserved from the data generation process [42]. The former approach requires fewer resources but more assumptions, and has been used with MNIST [35], CIFAR [32], and CelebA [39] datasets. These experimental setups rely on artificially inducing some of the characteristics of real federated datasets during the sampling process. Pachinko allocation based sampling method was proposed to generate more realistic heterogeneous partitions, but it requires a hierarchy of coarse labels such as present in CIFAR100 [46]. Yet, there is no clear way for measuring how realistic the splits are. In fact, federated data partitions in the wild are usually more heavy tailed than the artificial partitions previous work has used (see Section 4.2).

The latter approach relies on datasets generated from a collection of users, with the user identifiers preserved. Previous works have extensively used text datasets that have this property: Sentiment140 [17], Shakespeare [40], Reddit[8] and StackOverflow [6]. Realistic image datasets commonly used in the federated learning community are EMNIST [8], iNaturalist-User-120k and Landmarks-User-160k [25]. The landmarks dataset is the largest of them, with $164,172$ images, but has only 1262 users, making it ill-suited for large-scale federated learning, especially for private federated learning where large batch sizes are typically needed.

Meta-learning is a ML paradigm closely related to federated learning, hence requiring similar kinds of datasets. Popular image datasets for meta-learning include Mini-Imagenet [48], CUB-200-2011 [49] and Omniglot [34]. These datasets are relatively small and low-resolution, with either artificial task partitioning or easy tasks, e.g. the original model-agnostic meta-learning algorithm already achieves $99.9\%$ accuracy with 5-way 5-shot classification on Omniglot [16].

Testing a hypothesis with a standardized benchmark agreed upon by the research community is essential for systematically making progress in the field of machine learning. There are several benchmark suites that attempt to do this for federated learning: LEAF [8], FedML [21], OARF [26], FedGraphNN [20], FedCV [22], FedNLP [38] and FedScale [33]. FedScale proposes a benchmark for image classification on Flickr images, which is similar to FLAIR. This however is a multiclass dataset, where image-label pairs are constructed by cropping single objects from bounding box annotations; this results in many duplicate images with different labels because the bounding boxes commonly overlap. As explored more thoroughly in Section 4.2, FLAIR also has a more diverse set of classes and includes two levels of difficulty.

## 3 Preliminaries

**Federated learning** [40] enables training on users' data without collecting or storing the data on a centralized server. In each round of federated learning, the server samples a cohort of users and sends

the current model to the sampled users' devices. The sampled users train the model locally with SGD and share the gradient updates back to the server after local training. The server updates the global model, treating the aggregate of the per-user updates in lieu of a gradient estimate in an optimization algorithm such as SGD or Adam [31, 46].

**Differential Privacy.** Even though user data is not shared with the server in the federated setting, the shared gradient updates can still reveal sensitive information about user data [43, 53]. Differential privacy (DP) [14] can be used to provide a formal privacy guarantee to prevent such data leakage in the federated setting.

**Definition 1 (Differential privacy)** *A randomized mechanism $\mathcal{M} : \mathcal{D} \mapsto \mathcal{R}$ with a domain $\mathcal{D}$ and range $\mathcal{R}$ satisfies $(\epsilon, \delta)$-differential privacy if for any two adjacent datasets $d, d' \in \mathcal{D}$ and for any subset of outputs $S \subseteq \mathcal{R}$ it holds that $\Pr[M(d) \in S] \leq e^\epsilon \Pr[\mathcal{M}(d') \in S] + \delta$.*

In the context of DP federated learning [42], $\mathcal{D}$ is the set of all possible datasets with examples associated with users, range $\mathcal{R}$ is the set of all possible models, and two datasets $d, d'$ are adjacent if $d'$ can be formed by adding or removing all of the examples associated with a single user from $d$.

When a federated learning model is trained with DP, the model distribution is close to what it would be if a particular user did not participate in the training. Following prior works in DP-SGD [2] in the federated learning context [42], two modifications are made to the federated learning algorithm to provide a DP guarantee: 1) model updates from each user are clipped so that their $L_2$ norm is bounded, and 2) Gaussian noise is added to the aggregated model updates from all sampled users. For the purpose of privacy accounting, we assume that each cohort is formed by sampling each user uniformly and independently, and that this sample is hidden from the adversary.

# 4 FLAIR Dataset

## 4.1 Dataset collection

The initial set of images was curated with the Flickr API [2]. The corresponding Flickr user IDs were preserved so that the images were naturally grouped by users. All curated images are publicly shared by the Flickr users and permissively licensed (detailed in Appendix A).

**Filtering.** We enforce strict filtering criteria to remove images that may contain personally identifiable information (PII). We use a two stage filtering approach: 1) we apply a face detection model to automatically remove images with faces, and; 2) we rely on human annotators to filter the remaining images that contains PII. Specifically, two annotators were assigned for filtering each image where the first annotator flags whether an image contains PII and the second annotator validates the results. See Appendix A for detailed filtering guideline.

**Annotation.** The images from Flickr API were initially unlabeled. We annotated the images with the main objects present in the images using a taxonomy of 1,628 fine-grained classes. We also defined 17 coarse-grained classes in the taxonomy, where each fine-grained class is associated with a coarse-grained class. Similar to filtering, two annotators were assigned for labeling and validating each image. If there was an ambiguous object present in the image and the annotator could not tell which fine-grained label to assign, a coarse-grained label was added instead.

## 4.2 FLAIR dataset statistics

After filtering and annotation, the finalized FLAIR dataset contain 429,078 images from 51,414 Flickr users, with 17 coarse-grained labels and 1,628 fine-grained labels.

**User data statistics.** The number of images per user is significantly skewed, where the largest $2.3\%$ of users collectively have as many images as the bottom $97.7\%$ of users. The left of Figure 2 compares the quantity skew for FLAIR and other image classification benchmarks for federated learning. In the case of CIFAR, there is a straight line because there is no skew. The figure indicates that FLAIR has the second largest quantity skew, after iNaturalist-User-120k.

---

[2] https://www.flickr.com/services/api/

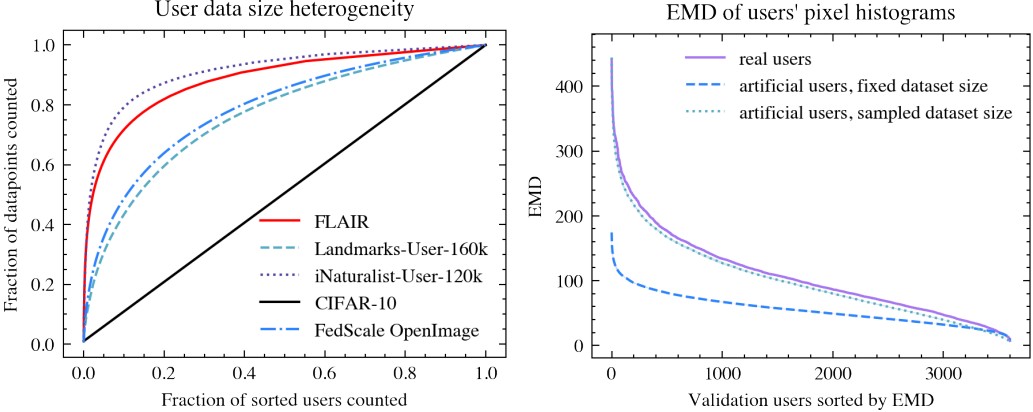

Figure 2: **Left:** Cumulative dataset length of users in descending order of quantity, normalized by number of users on x-axis and number of datapoints on y-axis. **Right:** Earth Mover's Distance (EMD) between users pixel histogram and the overall average pixel histogram from class *structure*. Blue dashed line is from simulating user splits with equal dataset size, green dotted line is simulating user splits by sampling from the real dataset size distribution, and purple solid line is the actual split.

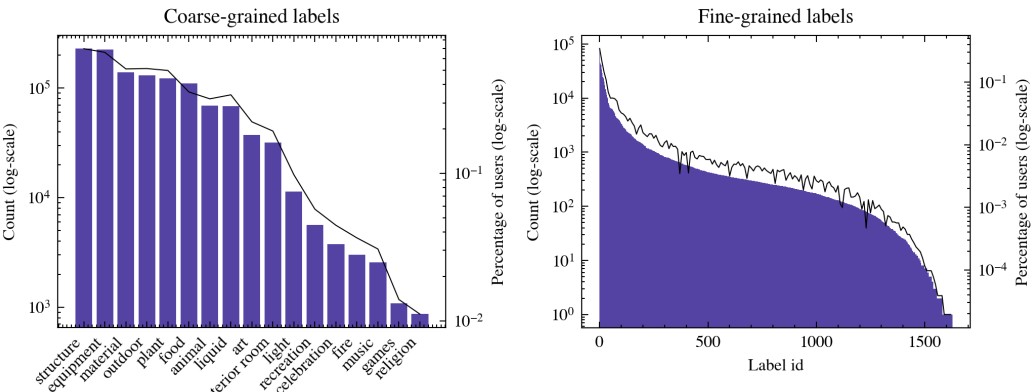

Figure 3: FLAIR label distribution for coarse-grained and fine-grained taxonomies. The bars are the counts of the labels from all images (left y-axis) and the curves are the percentages of the users who have the label (right y-axis).

To visualize the feature distribution skew in FLAIR, we show in Figure 2 (right) the Earth Mover's Distance (EMD) between the average pixel histogram of a user's images, to the population average pixel histogram. EMD is computed on the images from the most common label, *structure*, to remove any skew that the class imbalance might cause. The quantity skew also causes feature distribution skew (comparing blue dashed line to green dotted line), and the real non-iid partitioning slightly increases the skew compared to the average simulated non-iid partitioning (comparing green dotted line to purple solid line).

**Label statistics.** Figure 3 shows the total count of all labels across all users, revealing a significant class imbalance. The most common coarse-grained class, *structure*, occurs 228,923 times on a total of 87% of users. The least common coarse-grained class, *religion*, occurs 866 times on a total of 1.4% of users. The fine-grained labels similarly have a skewed distribution, with 1255 out of the 1628 classes being present on less than 0.1% of users.

**Dataset split.** For comparable and reproducible benchmarks on the FLAIR dataset, we provide a train-test split based on Flickr user IDs, such that the data of a particular user is only present in one of three partitions of the data. Out of 51,414 Flickr users, 80% are in the training, 10% in the validation set and 10% in the test set. There are 345,879 images in total in the training set, 39,239 in the validation set and 43,960 in the test set.

# 5 Experiments

## 5.1 Benchmark setups

**Learning settings.** We benchmark the FLAIR dataset in three different learning settings: centralized learning, non-private and private federated learning. Comparing these settings demonstrates how heterogeneity of the user data distribution and providing user privacy guarantees affect model convergence. In the centralized learning setting, training data is the union of images from all users in the training split and user ID is ignored.

**ML tasks and models.** As described in Section 4.1, the main objects in each image were annotated into coarse-grained and fine-grained taxonomies. We consider multi-label classification task on these two taxonomies, i.e. predicting if a class is present in an image for each class in the taxonomy.

We use a ResNet-18 [23] model for all benchmark experiments. The final classification layer is a 17-way logistic regression for the coarse-grained taxonomy and 1,628-way for the fine-grained. The model has more than 11M parameters in total. We consider both training from scratch (i.e. from a random initialization) and fine-tuning from a pretrained model. The pretrained ResNet-18 model is obtained from the Torchvision repository (version 0.12.0) [3] and was trained on the ImageNet dataset [11].

For models trained from scratch, we further replace all batch normalization (BN) [27] layers with group normalization [51] to avoid sharing the sensitive states in BN with the server in federated settings. For the pretrained ResNet-18 model, we freeze the BN states during fine-tuning and only update the scale and bias parameters.

**Evaluation metrics.** We use standard multi-label classification metrics for the benchmark, including precision (percentage of predicted objects that are actually in the images), recall (percentage of objects in the images found by the classifier), F1 score, and Average Precision (AP) score. We report overall (micro-averaged) metrics, obtained by averaging over all examples, and per-class (macro-averaged) metrics, obtained by taking the average over classes of the average over examples restricted to a specific class.

**Simulating large cohort noise-level with small cohort.** When training with DP, increasing cohort size $C$ will monotonically increase the signal-to-noise ratio (SNR) of the averaged noisy aggregates as the DP noise will be reduced by averaging. As we will show later in Section 5.2, the minimum SNR required for training large neural networks such as ResNet corresponds to a $C$ in the thousands, which is compute-intensive with current federated learning frameworks.

Following prior work [42], we simulate the SNR effect of a large cohort $C_{\mathrm{lg}}$ using a small cohort $C_{\mathrm{sm}}$ so that we can efficiently experiment with different noise-levels. Let $\sigma = \mathcal{M}(\cdot, C)$ be the noise multiplier calculated by moments accountant $\mathcal{M}$ [2] for cohort size $C$ and other privacy parameters. We use $C_{\mathrm{sm}}$ and noise multiplier $\sigma_{\mathrm{sm}}$ for experiments, where $\sigma_{\mathrm{sm}} = \frac{C_{\mathrm{sm}}}{C_{\mathrm{lg}}}\mathcal{M}(\cdot, C_{\mathrm{lg}})$. The noise applied to the averaged $C_{\mathrm{sm}}$ model updates has standard deviation $\frac{\sigma_{\mathrm{sm}}}{C_{\mathrm{sm}}} = \frac{\mathcal{M}(\cdot, C_{\mathrm{lg}})}{C_{\mathrm{lg}}}$, which is the same as if we are training with $C_{\mathrm{lg}}$ users.

**Hyperparameters.** For all experiments, we use FedAdam [31, 46] as the server-side optimizer. During training, each image is randomly cropped to size $224 \times 224$ and randomly flipped horizontally or vertically. During evaluation, each image is resized to $224 \times 224$. We performed a grid search on the hyperparameters and report the values that yield best performance on the validation set. See Appendix B.2 for hyperparameters grids.

For the centralized setting, we set the number of epochs to be 100 and the learning rate to be 5e-4 if training from scratch, and number of epochs to be 50 and learning rate to be 1e-4 when fine-tuning. We use a mini-batch size of 512.

For the federated learning setting, we train the model for 5,000 rounds with a cohort size of 200. We set the server learning rate to 0.1. Each sampled user trains the model locally with SGD for 2 epochs with local batch size set to 16 and local learning rate set to 0.1 when training from scratch and 0.01 when fine-tuning. We limit the maximum number of images for each user to be 512 and if a user has

---

[3] https://github.com/pytorch/vision

Table 1: FLAIR benchmark results on test set. All experiments are run for 5 times with different random seed, and both mean and standard deviation of metrics are reported. For setting, C, FL, PFL stands for centralized, federated and private federated learning; R and F stands for training from scratch and fine-tuning; 17 and 1628 are the number of classes in the coarse-grained and fine-grained taxonomies. For the metrics columns, C and O denotes whether the metrics are per-class or overall; AP denotes averaged precision; P denotes precision; R denotes recall; and F1 denotes F1 score.

| Setting | C-AP | C-P | C-R | C-F1 | O-AP | O-P | O-R | O-F1 |
|---|---|---|---|---|---|---|---|---|
| C-R-17 | $60.6_{\pm0.5}$ | $71.7_{\pm0.9}$ | $49.3_{\pm0.8}$ | $58.4_{\pm0.4}$ | $87.4_{\pm0.2}$ | $81.8_{\pm0.6}$ | $74.2_{\pm0.8}$ | $77.8_{\pm0.2}$ |
| FL-R-17 | $50.1_{\pm0.5}$ | $58.1_{\pm1.7}$ | $37.0_{\pm0.9}$ | $45.1_{\pm0.5}$ | $82.8_{\pm0.2}$ | $78.9_{\pm0.9}$ | $67.7_{\pm1.2}$ | $72.8_{\pm0.4}$ |
| PFL-R-17 | $29.2_{\pm0.1}$ | $29.1_{\pm1.1}$ | $18.6_{\pm0.5}$ | $22.7_{\pm0.7}$ | $63.6_{\pm0.2}$ | $67.9_{\pm0.5}$ | $45.7_{\pm1.1}$ | $54.6_{\pm0.7}$ |
| C-F-17 | $67.8_{\pm0.1}$ | $76.9_{\pm0.5}$ | $55.2_{\pm0.3}$ | $64.3_{\pm0.1}$ | $90.5_{\pm0.0}$ | $85.1_{\pm0.2}$ | $77.8_{\pm0.2}$ | $81.3_{\pm0.0}$ |
| FL-F-17 | $62.0_{\pm0.3}$ | $74.0_{\pm1.4}$ | $49.5_{\pm1.3}$ | $59.3_{\pm0.9}$ | $89.0_{\pm0.1}$ | $82.8_{\pm0.3}$ | $77.2_{\pm0.5}$ | $79.9_{\pm0.2}$ |
| PFL-F-17 | $43.8_{\pm0.2}$ | $49.1_{\pm2.6}$ | $32.1_{\pm0.4}$ | $38.8_{\pm0.6}$ | $80.2_{\pm0.1}$ | $77.8_{\pm0.3}$ | $63.8_{\pm0.7}$ | $70.1_{\pm0.3}$ |
| C-R-1628 | $14.6_{\pm0.2}$ | $25.6_{\pm0.7}$ | $6.7_{\pm0.1}$ | $10.6_{\pm0.2}$ | $42.8_{\pm0.3}$ | $66.9_{\pm0.9}$ | $25.5_{\pm0.7}$ | $36.9_{\pm0.6}$ |
| FL-R-1628 | $1.5_{\pm0.0}$ | $0.8_{\pm0.1}$ | $0.2_{\pm0.0}$ | $0.4_{\pm0.0}$ | $22.5_{\pm0.3}$ | $63.4_{\pm1.9}$ | $6.9_{\pm0.8}$ | $12.4_{\pm1.3}$ |
| PFL-R-1628 | $0.3_{\pm0.0}$ | $0.0_{\pm0.0}$ | $0.0_{\pm0.0}$ | $0.0_{\pm0.0}$ | $7.1_{\pm0.0}$ | $0.0_{\pm0.0}$ | $0.0_{\pm0.0}$ | $0.0_{\pm0.0}$ |
| C-F-1628 | $20.0_{\pm0.3}$ | $32.4_{\pm0.5}$ | $10.2_{\pm0.5}$ | $15.6_{\pm0.5}$ | $48.0_{\pm0.3}$ | $68.6_{\pm0.7}$ | $30.0_{\pm0.7}$ | $41.7_{\pm0.6}$ |
| FL-F-1628 | $2.0_{\pm0.1}$ | $1.6_{\pm0.1}$ | $0.4_{\pm0.0}$ | $0.6_{\pm0.0}$ | $27.0_{\pm0.4}$ | $64.5_{\pm1.6}$ | $10.5_{\pm0.6}$ | $18.0_{\pm0.9}$ |
| PFL-F-1628 | $0.5_{\pm0.0}$ | $0.2_{\pm0.0}$ | $0.0_{\pm0.0}$ | $0.0_{\pm0.0}$ | $12.3_{\pm0.2}$ | $53.9_{\pm5.0}$ | $0.2_{\pm0.1}$ | $0.4_{\pm0.2}$ |

more images, we randomly sample 512 images so that each sampled user maximally trains on 512 images for each local epoch.

For federated learning with differential privacy, we use $\epsilon = 2.0, \delta = N^{-1.1}$ where $N$ is the number of training users. We set the server learning rate to 0.02. We use an adaptive clipping algorithm [4] to tune the clipping bound, with the $L_2$ norm quantile set to 0.1. We use 200 users sampled per round to simulate the noise-level with a cohort size of 5,000, and we also analyze the effect of different cohort sizes in Section 5.2.

## 5.2 Results

Table 1 summarizes the benchmark results on the FLAIR test set. For the coarse-grained taxonomy, we observe that the performance gap between centralized and federated setting is about 20% on the per class metrics and 6% on the overall metrics if the models are trained from scratch. These gaps are reduced to 8% and 2% if models are fine-tuned from pretrained ResNet. When DP is applied, the per class metrics drop about 40% and overall metrics 24% from non-private federated learning if training from scratch. When fine-tuning with DP, the drop is less significant, about 30% for per class metrics and 10% for overall metrics.

For the fine-grained taxonomy, federated learning performance is much worse than the centralized baseline. The gaps are around 90% and 50% for per-class and overall metrics regardless whether the model is trained from scratch or started from a pretrained model. DP model has even worse performance compared to non-private one due to the extra noise introduced, which indicates long-tailed prediction tasks are especially hard in private federated learning setting due to the sparse label distribution among users.

**Per class results.** Figure 4 summarize averaged precision scores on FLAIR test set for each class in the coarse-grained taxonomy. The performances are different for different classes and there is a positive correlation between the frequency of the class and its performance. Noticeably, the gaps between classes are enlarged if models are trained with federated learning and DP. For instance, the gap between *recreation* and *outdoor* is about 68% in centralized setting while the gap increases to 81% in the federated setting and 96% in the federated setting with DP. In other words, the decrease in performance is worse for classes that are less frequent in federated learning, especially when DP is applied. This observation was also noted in prior works [7, 47].

**Per user results.** Figure 5 summarize the distribution of per user macro averaged precision scores from coarse taxonomy on FLAIR test set. As expected, the model performances vary among different users. There are a few users with much higher performances than majority for all training settings.

| Setting | struc-ture | equip-ment | mate-rial | out-door | plant | food | animal | liquid | art | interior room | light | recrea-tion | celeb-ration | fire | music | games | reli-gion |
|---|---|---|---|---|---|---|---|---|---|---|---|---|---|---|---|---|---|
| C-R | 90.1 | 92.6 | 66.8 | 95.3 | 92.9 | 94.8 | 86.5 | 79.0 | 42.1 | 64.6 | 34.3 | 33.1 | 37.5 | 64.5 | 17.5 | 19.0 | 19.7 |
| FL-R | 86.9 | 89.7 | 61.2 | 93.0 | 89.7 | 91.4 | 74.3 | 66.7 | 30.7 | 55.6 | 24.4 | 15.8 | 19.0 | 38.2 | 5.3 | 3.6 | 5.3 |
| PFL-R | 65.6 | 73.1 | 43.0 | 78.2 | 67.1 | 69.1 | 25.4 | 33.5 | 11.6 | 14.4 | 5.3 | 4.2 | 1.3 | 4.1 | 0.9 | 0.3 | 0.2 |
| C-F | 92.6 | 94.7 | 71.3 | 96.3 | 94.0 | 96.7 | 93.4 | 84.3 | 55.4 | 71.4 | 40.9 | 42.6 | 46.3 | 71.4 | 38.9 | 41.0 | 21.2 |
| FL-F | 91.5 | 93.9 | 69.4 | 95.7 | 93.1 | 95.8 | 91.3 | 80.5 | 49.7 | 69.1 | 37.2 | 29.0 | 37.9 | 65.6 | 19.7 | 24.7 | 9.8 |
| PFL-F | 84.0 | 88.6 | 55.6 | 89.7 | 86.0 | 90.6 | 77.1 | 55.0 | 26.0 | 51.4 | 15.2 | 6.3 | 4.0 | 13.0 | 1.8 | 0.4 | 0.5 |

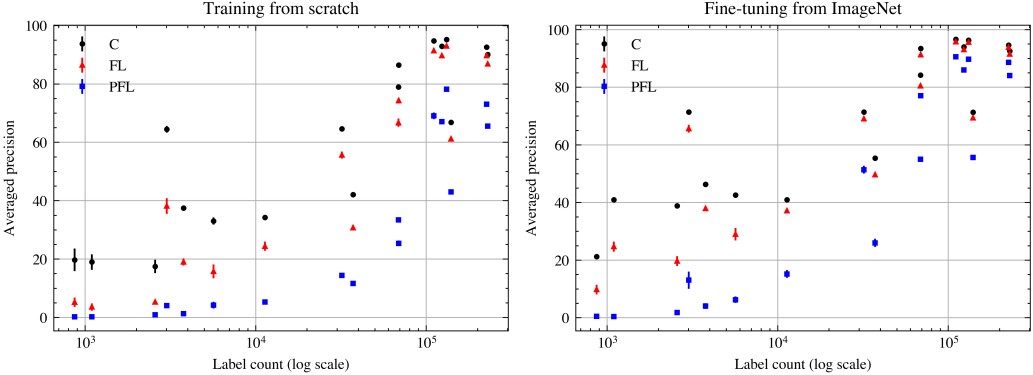

Figure 4: Averaged precision for each coarse-grained class. C, FL, PFL stands for centralized, federated and private federated learning. R and F stands for training from scratch and fine-tuning. Columns in the table are sorted by decreasing order of class frequency. For the figures, x-axis is the counts of each class in log scale and y-axis is the per-class AP on the validation set.

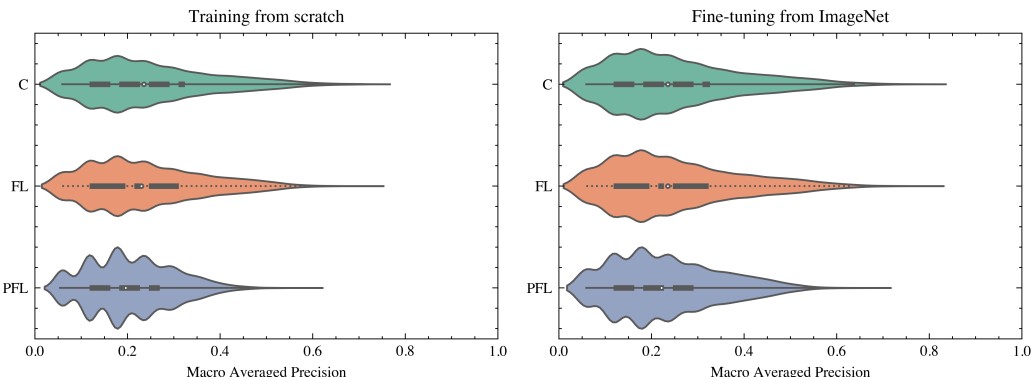

Figure 5: Violin plots for the distribution of per user macro averaged precision from coarse taxonomy on test set. C, FL, PFL stands for centralized, federated and private federated learning.

The distribution of scores are similar for centralized and federated learning setting while training with DP impacts the performances for almost all users.

**Effect of cohort size on PFL.** As described in Section 5.1, cohort size controls the noise-level of PFL, and thus we further examine the impact of cohort size on the performance of DP models. Figure 6 illustrates the per-class AP on the validation set in different rounds of PFL training. For both training from scratch and fine-tuning, increasing cohort size yields faster and better generalization.

# 6 Discussion

## 6.1 Research directions

**Imbalanced classes.** For the coarse-grained taxonomy, models performed differently on different classes and the performance is correlated to the frequency of the class. This difference is enlarged in federated learning, especially when DP is applied, indicating that the heterogeneity and DP noise

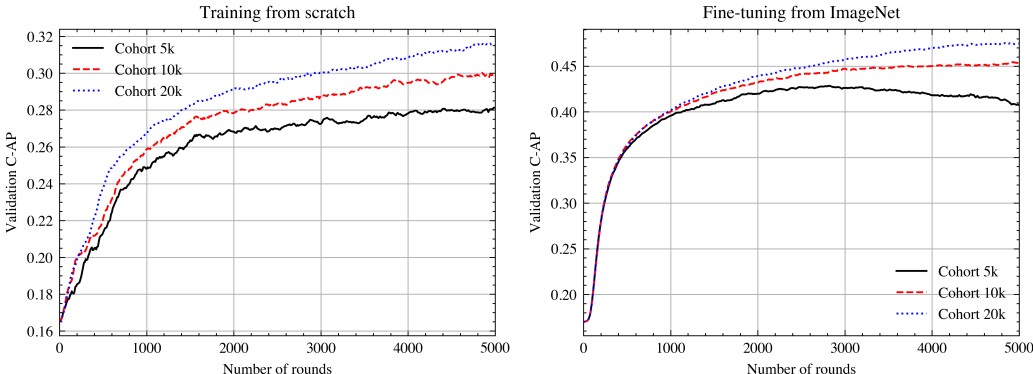

Figure 6: Effect of cohort size in PFL training. x-axis is the number of rounds of federated learning and y-axis is the per-class AP on the validation set.

worsened the imbalance problem. Given its heterogeneous nature, we believe FLAIR is a suitable dataset with which researchers can study the class imbalance problem in the distributed setting.

**Few-shot and zero-shot federated learning.** As demonstrated in Section 5.2, federated learning models perform worse on FLAIR fine-grained taxonomy compared to coarse-grained. Out of 1,628 fine-grained classes, 11 present in the validation and test dataset are unseen and 134 have less than 20 positive examples in the training set. Predicting these few-shot and zero-shot labels can be very difficult even in the centralized training setting. Indeed the signals for the tail classes in fine-grained taxonomy are extremely sparse and the sparsity is exacerbated in federated learning as the infrequent classes are concentrated in only a few users. Furthermore, DP exacerbates the performance of infrequent classes due to poor SNR of sparse gradients. We believe the long-tailed label distribution in FLAIR fosters research interests in few-shot and zero-shot learning in the private federated setting.

**Noise-robust and efficient federated learning with DP.** As shown in Figure 6, the larger the cohort size, the smaller the noise on the aggregated model updates and thus the better the model when trained with DP, especially for deep neural networks with tens of millions of parameters. Larger cohorts increase the latency of federated learning with DP and may become impractical when the number of iterations required to converge is also large. We believe the scale and complexity of FLAIR will inspire research in designing model architectures and optimization algorithms which are more robust to DP noise and also more efficient to train.

**Personalization.** Personalization in federated learning is an active research area as a single model is unlikely to generalize equally well among all users. Meta learning [15] and local adaptation [12, 52] are some of the attractive approaches for personalized federated learning. As many of the users in FLAIR have handful of images due to our strict filtering criterion, evaluating personalized federated learning algorithms on FLAIR can be challenging. We did not benchmark FLAIR with personalization in this work and leave it for future works.

**Advanced vision models.** As an initial benchmark, we only explored one model architecture, ResNet18. There are many more advanced architectures or pretrained models such as vision transformers [13], SimCLR [9], or CLIP [44] that we did not use for experiments. It is also an interesting research topic to search for the optimal model architectures in federated learning with DP.

**Advanced optimization algorithms.** There is a line of efforts that aims to tackle the heterogeneity in federated learning by more advanced optimization such as FedProx [36], Scaffold [30], Mime [29], and FedNova [50]. On the other hand, many recent works proposed optimization algorithms that improved upon DP-SGD [3, 5, 37]. As the main focus of this paper is to introduce the FLAIR dataset, we only benchmarked on federated learning algorithm, FedAdam [46]. We leave it for future works to benchmark FLAIR on the more advanced optimization algorithms in the DP and federated learning literature.

## 6.2 Limitations

Due to our strict filtering criteria, images with faces or identifiable human bodies are removed from FLAIR. Thus, FLAIR is not suitable for any facial recognition or person identification vision tasks. This filtering also reduced the size of the dataset, and may have impacted the distributions of the number of images per user. In addition, FLAIR does not contain bounding boxes or pixel-level annotations for the objects presented in the images and thus is not suitable for vision tasks such as object detection and image segmentation.

More generally, Federated Learning applications are diverse and various heterogeneity properties can vary a lot across applications. Any single dataset thus will not accurately represent all relevant properties of a specific application. Evaluating algorithms on a collection of datasets is thus important.

## 7 Conclusions

In this work, we presented FLAIR, a large-scale image dataset suitable for federated learning. We compared FLAIR with existing federated learning image datasets and discussed the advantages of FLAIR. We described how the images in FLAIR were curated and annotated. We provided reproducible benchmarks for centralized, federated and differentially private settings. We have open-sourced both the FLAIR dataset and the benchmark code for the community to use with the aim of in advancing the research in federated learning.

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
