# OpenReview forum: "FLAIR: Federated Learning Annotated Image Repository"
_NeurIPS.cc/2022/Track/Datasets_and_Benchmarks — NeurIPS 2022 Datasets and Benchmarks _

### Official Review · Reviewer_s4gW · 2022-07-19
**Useful large-scale dataset easily accessible accompanied by a well-writen article**

**Rating:** 9
**Confidence:** 5

**Strengths:**

- the authors provide a large-scale dataset for cross-device FL that is realistic, the download process is well documented in their github repository
- the authors convincingly show that the resulting heterogeneity coming from the natural split is interesting with respect to existing datasets and artificial splits
- the authors curate the dataset well using both automated and manual methods
- the article is extremely well-written and well-organized
- results on DP-FL and the influence of cohort-size (for sampling) are interesting

**Weaknesses:**

Major
- the authors do not provide code to reproduce their benchmarks => now they do
- there is no confidence intervals in the benchmark tables. One could argue that this is a dataset article not a benchmark article however if so the authors should change the wording so that benchmarks are not listed as contributions but as illustrative experiments. => they have been added


Minor
- I would suggest to plot the AP per class as a function of the count of images per class to supplement Table 2. This would better support the claims in the paragraph l231 to 238.
- The authors perform a server-size Adam and cite the FedOpt article, do the authors exactly follow FedAdam's algorithm ? If so the authors should say they use FedAdam and be more explicit as the federation of the Adam algorithm is not trivial.  If not the authors should describe the associated pseudo-code of the optimization.
- As authors consider local optimizations as doing 2 epochs for federated learning and add l210 "the maximum number of images per user is limited to 512" is 512 samples considered an epoch then for larger datasets ? This point needs clarification.
- Figure 4 does not print well in black and white please change the markers for the different cohort sizes.
 l19 "it is often impractical to upload data to servers" -> not precise enough, reword.
 l26 "a lot" -> many.
 l34 "feature and label distribution" -> features and labels distribution.
 l40 "the image domain suffers from a limited selection of large scale datasets with realistic partitions ..."-> I disagree with this statement it is a bit strong the image domain already has several FL benchmarks available for cross-device (LEAF, FedScale) that can be considered as large-scale, the authors mention them in the related works.
l75 "Pachinko allocation ... but it requires a hierarchy of coarse labels such as present in CIFAR100" -> isn't it the case for FLAIR with its coarse and fine-grained annotations ? If not explain why more clearly.
l144 Define "quantity skew" in the text in addition to explaining it in the figure.
l146 "earthmover distance (EMD)" -> Earth Mover Distance.
l149 the most common label is structure -> add that it is the most frequent for the coarse categories.
l159 "such that a user is only present in one of three partitions of the data" -> stratified by users.
l172 "presented" -> present.
l183 "percentage of objects in the images predicted" ->"percentage of objects in the images found by the classifier".
l184 average precision (AP) -> Average Precision.
l190 Can the authors provide a reference ?
l255 "strengthened" -> exacerbated.
l277 "changed"-> impacted.



**Additional Feedback:**

- How do the authors come up with 5000 rounds for FL ? Is there a link with the number of epochs (100) on pooled and the cohort size ? What is the reasoning ?
I will raise my notation significantly if authors include confidence intervals and benchmark code and answer satisfactorily to the rest of my questions.
=> I changed my rating from 7 to 9 overall I clearly believe this is a great article.

**Clarity:**

- The paper reads extremely well and contains few typos

**Correctness:**

The lack of confidence intervals does impact the quality of the benchmark but this is a dataset article.

=> the authors added multiple seeds results

**Documentation:**

The repository's instructions are clear and easy to read but only to download the datasets.
The licensing is adapted.

**Ethics:**

The authors did a good curation work and seem to be careful regarding nsfw content and potential identification of users.

**Relation To Prior Work:**

The related works section is sound although more works could be cited such as [1] that provide other cross-device datasets with natural splits using iNaturalists (https://www.tensorflow.org/federated/api_docs/python/tff/simulation/datasets/inaturalist).

[1] Keith Bonawitz, Hubert Eichner, Wolfgang Grieskamp, Dzmitry Huba, Alex Ingerman, Vladimir Ivanov, Chloe Kiddon, Jakub Konecnˇ y, Stefano Mazzocchi, Brendan McMahan, et al. `Towards federated learning at scale: System design. Proceedings of Machine Learning and
Systems, 1:374–388, 2019.



**Summary And Contributions:**

The authors provide a large-scale classification dataset for cross-device Federated Learning with two tracks; fine-grained classification 1628 classes and coarse classification with 17 classes.
This dataset has natural or "real-world" heterogeneity as it is scraped from Flickr and split by Flickr users.

In addition authors benchmark algorithms on those 2 on this dataset: pooled or central learning / regular FedAvg and central-server DP-FedAvg using both pretrained models and from-scratch trainings which makes, as expected, a big difference for DP-FL.



PS: The authors do not provide the NeurIPS checklist in the main article nor in the supplementary. I do not know if this is considered as a ground for rejection. The authors do provide an extensive checklist that, even if it differs from the one provided in the template, I would argue, is as relevant as the original in the Supplementary. I would advocate for not rejecting the article based on this.


PPS: The authors updated the code in the main branch of the repository recently after the supplementary deadline. My review is based on the version that was submitted for the deadline. I consider it highly unfair that the author that the authors push code to the main branch during the review period. Although working on the repository past the deadline is a good sign and exact rules were unclear regarding repositories, I consider it at least completely unwarranted. I did not take it into account in my review but would like the AC and other reviewers to know.

---

> ### Author Response · Authors · 2022-08-17
> **Response to Reviewer s4gW**
>
> Thank you for your thoughtful comments. Below we address the concerns you raised.
>
> > the authors do not provide code to reproduce their benchmarks
>
> Our repo included the code to reproduce the benchmarks (https://github.com/apple/ml-flair/tree/main/benchmark).
>
> > there is no confidence intervals in the benchmark tables
>
> Thank you for your suggestion. We have now added error bars and the averaged metrics to our tables.
>
> > I would suggest to plot the AP per class as a function of the count of images per class to supplement Table 2
>
> Thank you for your suggestion. We have now included Figure 4 which demonstrate the AP per class as a function of the count of images per class.
>
> > The authors perform a server-size Adam and cite the FedOpt article, do the authors exactly follow FedAdam's algorithm ?
>
> Yes, we followed exactly FedAdam and we have reworded that we use FedAdam more explicitly.
>
> > the maximum number of images per user is limited to 512" is 512 samples considered an epoch then for larger datasets ? This point needs clarification.
>
> Yes, each local epoch trains maximally 512 images. We have now added clarification on this point.
>
> > Figure 4 does not print well in black and white please change the markers for the different cohort sizes.
>
> Thank you for your suggestion. We have now included different line styles for Figure 6 (previously Figure 4).
>
> > How do the authors come up with 5000 rounds for FL ? Is there a link with the number of epochs (100) on pooled and the cohort size ? What is the reasoning ?
>
> These hyper-parameters are set similarly to prior works. Reddi et al [1] used FedAdam for 4,000 rounds on vision datasets CIFAR-10 and CIFAR-100 and thus we believe 5,000 is a reasonable ballpark number for the initial benchmark. The very first paper that proposed federated learning with differential privacy experimented with 5,000 cohort size [2]. For number of epochs in centralized training, the original ResNet paper trained ImageNet model for 600K iterations with batch size 256, or equivalently around 150 epochs [3]. Given that FLAIR has fewer number of images than ImageNet, we thus believe that 100 epoch is a reasonable ballpark number for the initial benchmark.
>
> **References:**
>
> [1] Reddi, S.J., Charles, Z., Zaheer, M., Garrett, Z., Rush, K., Konečný, J., Kumar, S. and McMahan, H.B., 2020, September. Adaptive Federated Optimization. In International Conference on Learning Representations.
>
> [2] McMahan, H.B., Ramage, D., Talwar, K. and Zhang, L., 2018, February. Learning Differentially Private Recurrent Language Models. In International Conference on Learning Representations.
>
> [3] He, K., Zhang, X., Ren, S. and Sun, J., 2016. Deep residual learning for image recognition. In Proceedings of the IEEE conference on computer vision and pattern recognition (pp. 770-778).

---

> > ### Comment · Reviewer_s4gW · 2022-08-19
> > **Acknowledgment of revisions**
> >
> > The authors complied to each of my requests. Their original results stand. I thus significantly improved my rating.
> > In addition most other reviewers seem to share my feelings.
> >
> > Furthermore, I think the weaknesses pointed out by reviewer 2BNV are minor as this contribution is focused on the dataset:
> > - Judging by the quality of the code, future pytorch users should be able to contribute pytorch wrappers easily to open the repository to a wider audience, same for model-personalization.
> > - I also agree with the authors that vertical FL doesn't apply at all in this context.

---

> > > ### Author Response · Authors · 2022-08-24
> > > **Response to Reviewer s4gW**
> > >
> > > We thank the reviewer for updating their score. If there are any additional questions, we would be happy to answer them.

---

### Official Review · Reviewer_eJgq · 2022-07-23
**Descriptions of the dataset and its limitations are detailed, and the paper is well written**

**Rating:** 8
**Confidence:** 3
**Correctness:** The dataset construction procedure is…
**Clarity:** There is an “e” missing in line 48

**Strengths:**

The dataset is collected from real Flickr profiles, which guarantees realistic user splits. Annotation of the images contains with both coarse and fine-grained labels, providing an area for further research on multi-class classification. The presented per-class statistics show the non-uniformity of the dataset.

The authors analyze models performance on the FLAIR dataset in three settings: centralized learning; federated learning; federated learning with central differential privacy. This allows to further understand the dataset structure and the challenges it presents.

**Weaknesses:**

As outlined by the authors, the dataset lacks human faces and bodies and therefore cannot be used for human-centered vision tasks.

The fine-grained classes are highly imbalanced and lead to pure classification results. It is not clear if these labels will be actively used during further research.

**Additional Feedback:**

.

**Documentation:**

The repository for the dataset is well-documented, and the data is available for download.

**Ethics:**

The dataset consists of such Flickr images, which are publicly shared and permissively licensed.

**Relation To Prior Work:**

The authors observe existing datasets and studies, and describe drawbacks of previous datasets which the proposed dataset aims to cover.

**Summary And Contributions:**

The paper presents FLAIR, a large-scale image dataset for federated learning. The images in the dataset are gathered and filtered in such a fashion that no faces are present to avoid any personally identifiable information in the dataset
This dataset contains more images and user IDs than previous datasets, which contributes to the federated learning research.
The authors discuss advantages and limitations of the dataset and outline directions for future research.

---

> ### Author Response · Authors · 2022-08-17
> **Response to Reviewer eJgq**
>
> Thank you for your thoughtful comments. Below we address the concerns you raised.
>
> > The fine-grained classes are highly imbalanced and lead to pure classification results. It is not clear if these labels will be actively used during further research.
>
> We agree that learning with highly imbalanced and long-tailed label distribution is challenging, and our initial benchmark showed that such problem is even harder in the private federated learning setting. Many existing works [1, 2, 3, 4] in the vision domain attempted to tackle this problem in the centralized training setting. However, this is less explored in the federated setting and we believe that highly imbalanced fine-grained classes make FLAIR an ideal dataset to study such a challenging problem.
>
> **References:**
>
> [1] Liu, Z., Miao, Z., Zhan, X., Wang, J., Gong, B. and Yu, S.X., 2019. Large-scale long-tailed recognition in an open world. In Proceedings of the IEEE/CVF Conference on Computer Vision and Pattern Recognition (pp. 2537-2546).
>
> [2] Kang, B., Xie, S., Rohrbach, M., Yan, Z., Gordo, A., Feng, J. and Kalantidis, Y., 2020, September. Decoupling Representation and Classifier for Long-Tailed Recognition. In International Conference on Learning Representations.
>
> [3] Tang, K., Huang, J. and Zhang, H., 2020. Long-tailed classification by keeping the good and removing the bad momentum causal effect. Advances in Neural Information Processing Systems, 33, pp.1513-1524.
>
> [4] Tan, J., Wang, C., Li, B., Li, Q., Ouyang, W., Yin, C. and Yan, J., 2020. Equalization loss for long-tailed object recognition. In Proceedings of the IEEE/CVF conference on computer vision and pattern recognition (pp. 11662-11671).

---

> > ### Author Response · Authors · 2022-08-24
> > **Response to Reviewer eJgq**
> >
> > We thank the reviewer again for their feedback and hope we have addressed their questions/concerns. If there are any additional questions, we would be happy to answer them.

---

### Official Review · Reviewer_eA2n · 2022-07-23
**A large-scale image classiﬁcation dataset with comprehensive non-IID characteristics for benchmarking federated learning (FL) in various FL settings.**

**Rating:** 7
**Confidence:** 4
**Clarity:** The paper is well written.

**Strengths:**

1.	FLAIR has a large data scale and natural user segmentation, and considers many common non-IID characteristics.
2.	The proposed two levels of difﬁculty, the easier 17 coarse-grained classes and the harder 1,628 ﬁne-grained classes, can meet more diverse FL settings compared to other federated datasets.

**Weaknesses:**

1.	As mentioned in the discussion, out of 1,628 ﬁne-grained classes, 11 present in the validation and test dataset are unseen in the training set. When doing normal FL tasks (tasks other than few-shot and zero-shot FL), will this part of data affect the judgment of performance?

**Additional Feedback:**

FLAIR may be more influential by improving the annotation of data sets to object detection or image segmentation tasks.

**Correctness:**

Yes. The construction of the proposed FLAIR is reasonable and meets the needs of most FL tasks.

**Documentation:**

Yes.

**Ethics:**

After reading the paper, I think it is unlikely to have a negative impact on society.

**Relation To Prior Work:**

The authors discuss the relationship between FLAIR and other federated datasets in detail in Related Work. Compared with prior works, FLAIR has more realistic heterogeneous partitions and larger data scale (including the number of users and images). Compared with the latest FedScale, FLAIR has a more diverse set of classes and includes two levels of difﬁculty.

**Summary And Contributions:**

This paper builds a large-scale multi-label image classiﬁcation dataset FLAIR (including 429,078 images from 51,414 users, with 17 coarse-grained labels and 1,628 ﬁne-grained labels) to benchmark federated learning algorithms and models. FLAIR collects the images from Flickr and ﬁlters out the personally identiﬁable information (PII). To be more realistic, FLAIR considers several important non-IID challenges in federated learning and introduces imbalanced partitions, feature/label distribution skew, conditional feature distribution skew and label shifts into the dataset. This paper also provides the performance evaluation in some typical FL settings on FLAIR.

---

> ### Author Response · Authors · 2022-08-17
> **Response to Reviewer eA2n**
>
> Thank you for your thoughtful comments. Below we address the concerns you raised.
>
> > When doing normal FL tasks (tasks other than few-shot and zero-shot FL), will this part of data affect the judgment of performance?
>
> This part of data only consists a small set of all the evaluation datapoints and is unlikely affect the judgment of performance.
>
> > FLAIR may be more influential by improving the annotation of data sets to object detection or image segmentation tasks.
>
> Thank you for pointing this out, we have now acknowledged this in the limitation section (6.2).

---

> > ### Author Response · Authors · 2022-08-24
> > **Response to Reviewer eA2n**
> >
> > We thank the reviewer again for their feedback and hope we have addressed their questions/concerns. If there are any additional questions, we would be happy to answer them.

---

### Official Review · Reviewer_2BNV · 2022-07-28
**Good contribution on large-scale cross-device FL benchmark, but only useable with TensorFlow**

**Rating:** 6
**Confidence:** 5
**Clarity:** Paper is very well written.

**Strengths:**

1. Provides a large-scale FL dataset for multi-class image classification problem **with real partitions**.
2. Provides the dataset with a good motivation and mentions the challenges in datasets. Listing the skews in the dataset is very useful. Allows performing vertical and horizontal FL.
3. Benchmarks for private federated learning with large cohorts.
4. Nice demonstration on the effect of the cohort size over accuracy.


**Weaknesses:**

1. Proposed benchmark is only for vision tasks that do not include federated NLP tasks such as federated keyword prediction.
2. It would be great if you could demonstrate the performance of personalized FL algorithms, which is important in this setting as it is also mentioned by the authors.
3. It seems FLAIR only supports TensorFlow, which I think is a huge disadvantage for PyTorch users.
4. Dataset could support vertical FL, but it seems experimentation on vertical FL is weak or none.

**Additional Feedback:**

The only thing I could not find is how PyTorch users would use FLAIR? This is a big weakness of the paper. Also, it would be great to simulate benchmark results on a a vertical FL setting where the label space may differ.

**Correctness:**

Evaluation methods and experiment design are appropriate and performed correctly.

**Documentation:**

There is sufficient detail to support reproducibility. The only thing that I could not find is how would PyTorch users use FLAIR? This is a big weakness of the paper.

**Ethics:**

There are no issues with ethics guidelines.

**Relation To Prior Work:**

Section 2 explains the differences clearly. Moreover, it also includes popular benchmarks. However, some are missing. It would be nice if you cite the following papers:

[1] He, Chaoyang, et al. "Fedgraphnn: A federated learning system and benchmark for graph neural networks." arXiv preprint arXiv:2104.07145 (2021).

[2] Lin, Bill Yuchen, et al. "Fednlp: Benchmarking federated learning methods for natural language processing tasks." arXiv preprint arXiv:2104.08815 (2021).

[3] He, Chaoyang, et al. "Fedcv: a federated learning framework for diverse computer vision tasks." arXiv preprint arXiv:2111.11066 (2021).

**Summary And Contributions:**

This paper proposes a new benchmark for large-scale cross-device federated multi-class image classification problem.

---

> ### Author Response · Authors · 2022-08-17
> **Response to Reviewer 2BNV**
>
> Thank you for your thoughtful comments. Below we address the concerns you raised.
>
> > Proposed benchmark is only for vision tasks that do not include federated NLP tasks.
>
> Our main focus of this paper is to introduce the FLAIR dataset rather than providing a comprehensive benchmark for different data domains. We have discussed in Section 1 and 2 that there are many existing federated datasets in text domain already, and why a high quality federated dataset is needed in the vision domain.
>
> > It would be great if you could demonstrate the performance of personalized FL algorithms, which is important in this setting as it is also mentioned by the authors.
>
> Thank you for your suggestion. We have included more discussion on personalized FL on FLAIR in Section 6.
>
> > The only thing I could not find is how PyTorch users would use FLAIR? This is a big weakness of the paper
>
> FLAIR dataset supports Pytorch and any other machine learning framework. We provided the raw dataset as well as preprocessing script (https://github.com/apple/ml-flair/blob/main/prepare_dataset.py) that transform the raw dataset into HDF5 format which is compatible to any framework and not just TensorFlow. Pytorch (and other framework) users can use FLAIR from the processed HDF5 dataset or the raw dataset.
>
> > It would be great to simulate benchmark results on a a vertical FL setting where the label space may differ
>
> To the best of our understanding, vertical FL refers to the scenario that participants hold different features while share the same sample ID space [1, 2, 3]. FLAIR is horizontally split by user ID and thus not suitable for vertical FL setting.
>
> > Some citations are missing.
>
> Thank you for pointing us to these papers, and we have now cited them in Section 2.
>
> **References:**
>
> [1] Skillicorn, D.B. and McConnell, S.M., 2008. Distributed prediction from vertically partitioned data. Journal of Parallel and Distributed computing, 68(1), pp.16-36.
>
> [2] Hardy, S., Henecka, W., Ivey-Law, H., Nock, R., Patrini, G., Smith, G. and Thorne, B., 2017. Private federated learning on vertically partitioned data via entity resolution and additively homomorphic encryption. arXiv preprint arXiv:1711.10677.
>
> [3] Yang, Q., Liu, Y., Chen, T. and Tong, Y., 2019. Federated machine learning: Concept and applications. ACM Transactions on Intelligent Systems and Technology (TIST), 10(2), pp.1-19.

---

> > ### Author Response · Authors · 2022-08-24
> > **Response to Reviewer 2BNV**
> >
> > We thank the reviewer again for their feedback and hope we have addressed their questions/concerns. If there are any additional questions, we would be happy to answer them.

---

### Official Review · Reviewer_XSNs · 2022-07-28
**Good paper addressing a real need for cross-device Fl**

**Rating:** 8
**Confidence:** 3

**Strengths:**

- This paper is extremely relevant to the cross-device community: for images, researchers mainly rely on synthetic splits of classic image datasets (MNIST, CIFAR10, sometimes ImageNet), which probably does not reflect the true challenges of this setting. This work is really needed, providing an open gamefield to the community.
- The code is lightweight and seems easily understandable. Although for storage reasons I did not have time to download the dataset, it seems fairly easy with the provided code and documentation.

**Weaknesses:**

- Only 1 DL model (Resnet-18) and 1 FL algorithm (FedAdam) are benchmarked by the authors
- In the proposed benchmark Sec 5.1, the authors mainly propose performance-related average metrics L182-187 (either micro-averaged or macro-averaged). Although such average metrics can be useful to compare different methods, looking at distributions of these metrics (e.g. box-plots) across users or classes could bring some richness in the benchmark. Note that researchers are not prevented from doing it by the benchmark.

**Additional Feedback:**

Minor comments
- The authors indicate L177 that the pretrained ResNet-18 model used was fetched from torchvision. However, the exact version of the software is not precised in the article and in the codebase. When checking in detail the code base, one sees that the authors re-implemented this network in tensorflow based on the pytorch implementation. There are no visible tests on checking the correctness of this re-implementation. At the very least, I would recommend adding the exact torchvision version that was used.

Minor presentation suggestions
- This reviewer is colorblind, and I thank the authors for making all figures readable thanks to dashed/dotted lines. To make the presentation even simpler to grasp, in Figure 2 left's caption as well as in the text L151-152, I would recommend writing "Dotted lines" instead of "Blue lines", and same for the others: this would make it much faster to grasp the authors' comments of the figure.
- L184, a "that" or "which" might be missing between "images" and "are predicted"
- in Figure 3, maybe adding the number or percentage of users in addition to the raw count of examples on a right axis could convey more information

**Clarity:**

The paper is very well written. I have added some very minor suggestions below to improve it further.

**Correctness:**

- The construction of both the dataset and the benchmark are sound to me.
- I have a minor concern with respect to the claim the claim regarding the potential presence of label shifts in the dataset (L61-63): although the authors clearly say there are no significant discrepancies, they hint at potential differences due to the multiplicity of human annotators. No experiment backs this claim. Further, if I well understood the paper, L136 indicates that only 2 annotators were used and worked in tandem, one annotating and the other one validating. Therefore, the potentiality hinted at by the authors does not seem very realistic.

**Documentation:**

- There is a readme in the github repository with instructions on installation and reproducing the experiments. Although it seems very clear to me, I did not get the chance to actually run the instructions due to storage constraints.

**Ethics:**

No ethic concern: the authors removed images with faces from the dataset through manual annotation.

**Relation To Prior Work:**

The paper cites well previous cross-device federated learning datasets.

**Summary And Contributions:**

This paper introduces a novel cross-device federated dataset, FLAIR (Federated Learning Annotated Image Repository), with associated download and benchmark code stored in github. The dataset consists in 429k images from 51k individual Flickr users. The task consists in classification, with 2 levels of granularity for the classes: 17 coarse and ~1.6k fine classes. After a description of the dataset, the authors perform a benchmark in 3 settings, for each class granularity level: centralized training, federated training (with FedAdam) and differentially private federated training. The results show the difficulty of the task, with centralizing training getting better results than FL, which is in turn better than DP-FL, thereby providing an interesting challenge to the community.

The main contribution of the paper lies in the creation of the dataset, which is original by its scale. Only 1 other dataset has a number of images of the same magnitude, but with 1 order less of users.

---

> ### Author Response · Authors · 2022-08-17
> **Response to Reviewer XSNs**
>
> Thank you for your thoughtful comments. Below we address the concerns you raised.
>
> > Only 1 DL model (Resnet-18) and 1 FL algorithm (FedAdam) are benchmarked by the authors
>
> Our main focus of this paper is to introduce the FLAIR dataset. We acknowledged that our benchmark is initial and not comprehensive in terms of evaluating DL models in Section 6. We have now stressed that in the Section 6 as well for evaluating FL algorithms.
>
> > distributions of these metrics (e.g. box-plots) across users or classes could bring some richness in the benchmark
>
> Thank you for the suggestion. We added the corresponding figures (Figure 4 and 5) showing the distribution of metrics across classes and users in Section 5.
>
>  > minor concern with respect to the claim the claim regarding the potential presence of label shifts in the dataset (L61-63)
>
> We have removed this claim about potential label shifts in the introduction.
>
> > ResNet-18 model: the exact version of the software is not precised in the article and in the codebase
>
> Thank you for the suggestion, we have provided the exact pytorch vision version in the article and codebase.
>
> > in Figure 3, maybe adding the number or percentage of users in addition to the raw count of examples on a right axis could convey more information
>
> Thank you for the suggestion, we have added the percentage of users in right axis of Figure 3.

---

> > ### Author Response · Authors · 2022-08-24
> > **Response to Reviewer XSNs**
> >
> > We thank the reviewer again for their feedback and hope we have addressed their questions/concerns. If there are any additional questions, we would be happy to answer them.

---

> > ### Comment · Reviewer_XSNs · 2022-08-26
> > **Thanks for your response**
> >
> > I would like to thank the authors for their response and congratulate them for the rebuttal work. All my concerns have been addressed. I am convinced this is a strong paper which should be accepted for publication.

---

### Meta-Review · Area_Chair_An6M · 2022-09-09

**Recommendation:** Accept
**Confidence:** 5

**Metareview:**

The authors provide a benchmark for federated learning.

The commonly mentioned strengths:
- there is a big need in the community for this type of work
- easily accessible code
- various levels of complexity with different granularity

Regarding the weaknesses, I see no showstoppers
- some feedback about the experiments done on the datasets -> this point comes back consistently among all reviewers, in terms of the amount, settings and reproducibility.. The authors are adviced to address this in future work, to keep momentum for the use of the community on this dataset
- there was some discussion about tensorflow and pytorch, but based on the discussion both seem supported
- some settings / image types could be better supported

None of these weaknesses seem serious, and quite frankly, we can not expect one dataset to cover all possible settings. Since the reviewers are in agreement about the quality of this work, I recommend them for an oral presentation.

---

### Decision · Program_Chairs · 2022-09-16

Accept